# Feature Dynamics as Implicit Data Augmentation: A Depth-Decomposed View on Deep Neural Network Generalization

## Abstract

Why do deep networks generalize well? In contrast to classical generalization theory, we approach this fundamental question by examining not only inputs and outputs, but the evolution of internal features. Our study suggests a phenomenon of temporal consistency in which predictions remain stable when shallow features from earlier checkpoints combine with deeper features from later ones. This stability is not a trivial convergence artifact. It acts as a form of implicit, structured augmentation that supports generalization. We demonstrate that temporal consistency extends to unseen and corrupted data but disappears when the training supervision lacks semantic structure (e.g., random labels). Statistical tests further reveal that stochastic gradient descent (SGD) injects anisotropic noise aligned with a few principal directions, reinforcing its role as a source of structured variability. Together, these findings suggest a conceptual perspective that links feature dynamics to generalization, pointing toward future work on practical surrogates for measuring temporal feature evolution.

## Keywords

Generalization, deep learning, feature dynamics, implicit bias, robustness

## 1 Introduction

Modern deep networks generalize well even in regimes where classical theory predicts overfitting. Explaining this gap is not only theoretically valuable but also central to robustness, sample efficiency, and principled training design. Capacity-based accounts (VC dimension, Rademacher complexity, uniform convergence (Vapnik, 1998; Bartlett & Mendelson, 2002; Bartlett et al., 2017; Neyshabur et al., 2017)) often become vacuous in highly overparameterized settings: they bound worst-case hypothesis classes but miss the inductive biases of optimization. Stability-based analyses and implicit bias perspectives (e.g., flat minima Wu et al. (2017); Wu & Su (2023)) better reflect training but remain in parameter space, leaving open how representations evolve to support generalization. Phenomena such as neural collapse (Papyan et al., 2020) capture elegant late-stage geometry but not the dynamics before collapse or the reuse of intermediate features. Meanwhile, data augmentation and invariance learning are known to improve generalization but they are typically handcrafted. This raises a natural question: Could augmentation-like mechanisms emerge organically from the training process itself?

We revisit generalization through a depth-decomposed lens centered on representation dynamics. In writing $f = f_{[d+1:n]} \circ f_{[1:d]}$, we examine how the deep classifier interacts with shallow features as they evolve during training. This motivates the hypothesis that the shallow network acts as an implicit augmenter for the deep network: As optimization proceeds, $f_{[1:d]}(x)$ generates semantically coherent feature variants. If $f_{[d+1:n]}$ can classify features drawn from different training times, training itself implicitly provides structured, temporally varying augmentations. Across architectures and datasets, three regularities consistently emerge: (i) memory and forgetting: later classifiers remain predictive on earlier features within a temporal window that gradually extends during training; (ii) transferability: earlier classifiers can still process later features when the shift is moderate, reflecting structured rather than arbitrary drift; and (iii) induction: feature trajectories align with final decision regions.

Two diagnostics connect temporal consistency to generalization. First, even after training metrics have stabilized, parameters and features continue to move at a non-negligible scale, indicating that the memory window is not a trivial convergence artifact or fully developed neural collapse. Second, when semantic structure is destroyed (random labels or heavy corruption), the consistency distribution collapses, and composite models lose predictivity. Thus, temporal variability must be semantically structured to benefit generalization.

To probe the mechanism, we measure one-step perturbations for the stochastic gradient descent (SGD). The resulting noise covariance is clearly anisotropic, with variance concentrated in a few leading directions. Isotropy tests consistently reject the spherical null, confirming that SGD injects structured rather than isotropic variability. Moreover, the strength of this anisotropy correlates with temporal consistency and memory-window length, linking SGD-induced noise to generalization.

Finally, we outline a conceptual bridge: if a classifier remains temporally consistent within a window, class-wise generalization gaps can be related to distances between temporally augmented feature distributions and their clean counterparts. While this TV-style formulation is not yet computable, it frames temporal consistency as a mechanism for generalization and points toward tractable surrogates such as MMD or Wasserstein distances.

The intuition of this study stems from a clean observation about ResNets Gai & Zhang (2021)Li & Papyan (2024), later extended to transformers Aubry et al. (2025), that the trajectories of training data points in the forward propagation of a deep neural network with residual connections converge to straight lines at the end of the training process. A natural question arises: How do deep networks generalize to the region surrounding these lines? In this paper, we address this question by investigating the 'feature dynamic' in the training process.

Our contributions are threefold:

- New lens on generalization. We introduce a depth-decomposed framing where feature dynamics act as implicit, structured augmentations. This perspective is operationalized through composite networks and temporal consistency metrics.
- Robust empirical phenomena. Across datasets and architectures, we show that temporal consistency is tightly coupled with generalization: it holds on unseen and corrupted data but collapses under random labels, highlighting its semantic dependence.
- Mechanistic link to SGD. Through perturbation analysis, we demonstrate that SGD injects anisotropic noise aligned with feature dynamics, and we provide a conceptual bridge connecting this structured variability to generalization, suggesting computable surrogates such as MMD or Wasserstein distances.

## 2 RELATED WORKS

### 2.1 GENERALIZATION

The generalization ability of neural networks remains a central focus in deep learning research (Neyshabur et al., 2014; Zhang et al., 2016). Despite having more parameters than training data, these models often generalize remarkably well. Traditional learning theories, such as VC-dimension (Vapnik, 1998) and Rademacher complexity (Bartlett & Mendelson, 2002), struggle to explain this behavior, particularly in over-parameterized and non-convex settings typical of deep learning. Consequently, new theoretical frameworks have emerged to better account for the generalization of neural networks. But the contribution of depth to generalization has not been fully deciphered.

1. Optimization-based generalization research: A substantial body of research examines how optimization influences neural network generalization. Hardt et al. (2016) demonstrated that SGD implicitly regularizes models, helping to prevent overfitting. Wu et al. (2017) analyzed the loss landscape geometry, finding that optimization often converges to flat minima, which correlate with better generalization. Fu et al. (2023) further linked generalization to the complexity of the learning trajectory. However, the stochastic perturbations introduced by SGD remain inadequately characterized mathematically across parameters from different layers.

2. Model capacity and complexity-based generalization research: Zhang et al. (2016) showed that neural networks can generalize well even when their parameter count far exceeds the number of

training samples, particularly on large datasets. This challenges traditional learning theory, which suggests that increased model complexity leads to overfitting. Moreover, classical complexity measures such as VC-dimension, Rademacher complexity, and related methods (Bartlett et al., 2017; Neyshabur et al., 2017) fail to fully account for the strong generalization observed in deep neural networks, especially on large-scale datasets (Nagarajan & Kolter, 2019). For two-layer networks, Ma et al. (2022) used Barron space to characterize the associated function space. However, for deep networks, defining their corresponding function spaces remains an open problem.

3. Phenomenon-driven generalization research: Prior studies have examined generalization through empirical phenomena, e.g., distributional generalization (Nakkiran & Bansal, 2020) and the generalization disagreement equality Jiang et al. (2021). These works focus on cross-model or input–output stability. In contrast, we analyze intra-model feature dynamics: the evolving shallow features of a single network act as structured, temporally varying augmentations for deeper layers, offering a new lens distinct from prior cross-model comparisons.

## 3 SETUP & NOTATION

We write the network as a composition $f = f_{[d+1:n]} \circ f_{[1:d]}$, where the shallow subnetwork $f_{[1:d]}$ acts as a feature extractor and the deep subnetwork $f_{[d+1:n]}$ serves as a classifier. Given two checkpoints $t_1 \le t_2$, define the **composite networks** $f_{[d+1:n]}(\theta_{t_2}) \circ f_{[1:d]}(\theta_{t_1})$ and $f_{[d+1:n]}(\theta_{t_1}) \circ f_{[1:d]}(\theta_{t_2})$ (Figure 1).

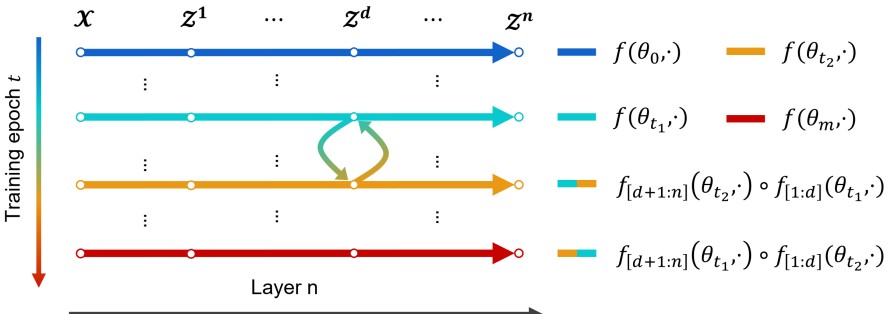

Figure 1: $\mathcal{Z}^k$ represents the $k$-th latent space. Each row represents a network at a specific epoch with colors indicating training time. Trajectories with two colors correspond to our designed composite networks.

In continuous time, the dynamics of gradient-based optimization can be described by differential equations. Let $\theta_t \in \mathbb{R}^d$ denote the parameters at time $t$, and $L(\theta)$ the loss. The gradient flow is:

$$\frac{d\theta_t}{dt} = -\nabla_\theta L(\theta_t)$$

while the stochastic gradient descent (SGD) dynamics include an additional noise term Chen et al. (2022); Chaudhari & Soatto (2018):

$$d\theta_t = -\nabla_\theta L(\theta_t)\, dt + \sqrt{\Sigma}\, dB_t$$

where $\Sigma$ is the noise covariance matrix and $B_t \in \mathbb{R}^d$ a Brownian motion. This parameter noise propagates to hidden features. For input $x$, the first-layer representation is $z_t^1 = f_1(\theta_t^1, x) \in \mathbb{R}^{w_1}$, with $\theta_t^1$ the first-layer parameters and $w_1$ its width. By Itô's lemma,

$$dz_{t,k}^1 = \left( -\frac{\partial f_{1,k}}{\partial \theta} \nabla_\theta L + \tfrac{1}{2}\mathrm{tr}\big(\nabla_\theta^2 f_{1,k}(\theta_t^1, x)\, \Sigma\big) \right) dt + \frac{\partial f_{1,k}}{\partial \theta} \sqrt{\Sigma}\, dB_t,$$

for each component $z_{t,k}^1$. Hence feature dynamics consist of both deterministic gradient-driven evolution and stochastic fluctuations injected by SGD. As a result, the hidden feature $z_t(x)$ does not remain a single deterministic point but evolves into a distribution, which we interpret as an implicit form of data augmentation arising from the stochastic feature dynamics (Appendix A for

detail). Conventional data augmentation, based on hand-crafted transformations or external generative models, improves robustness to repeated patterns of the same augmentation (Quiroga et al., 2018; Hendrycks et al., 2019). In contrast, we consider a naturally emerging augmentation mechanism internal to the network: shallow layers act as implicit augmenters. Accordingly, we ask whether deeper layers exploit this effect by learning to be robust to features extracted by shallow layers across different training epochs.

## 4 PHENOMENA: FEATURE EVOLUTION AS STRUCTURED AUGMENTATION

Building on Section 3, where we showed that hidden features evolve as distributions under SGD dynamics, we now ask whether deeper classifiers actually exploit this temporal variability. Using composite models that pair shallow layers from one checkpoint with deeper layers from another, we find that accuracy and consistency remain high within a temporal window. This shows that as shallow features drift over training, deep classifiers continue to process them reliably, indicating that the network has learned these evolving features, thereby supporting the data-augmentation hypothesis.

**Reproducible phenomena.** We now empirically examine the three phenomena introduced in Section 1—memory and forgetting, transferability, and induction. Across datasets and architectures, three patterns consistently emerge: later classifiers remain predictive on earlier features within a temporal window, earlier classifiers can process moderately shifted later features, and feature trajectories align with the final decision boundaries. Together they demonstrate that networks learn on evolving feature distributions, and that these features implicitly shape the classifier's geometry.

### 4.1 EXPERIMENTAL SETUP

To test these phenomena, we use composite networks—formed by pairing shallow layers from one epoch with deep layers from another (as introduced in Sec. 3).

**Experiment 1.** Memory and forgetting. To test memory, we pair shallow layers from an earlier epoch with deep layers from a later epoch and evaluate the resulting composite network. This reveals whether later classifiers can still recognize features produced at earlier stages, and how performance decays as the temporal gap widens.

**Experiment 2.** Transferability. To test transferability, we reverse the roles of early and late checkpoints, asking whether earlier classifiers can still process features extracted at later stages of training.

**Experiment 3.** Feature trajectories and classification regions. To examine induction, we visualize intermediate features across epochs and compare their trajectories with the network's final decision boundaries. Low-dimensional datasets allow direct plotting, while higher-dimensional ones (e.g., CIFAR-10) are projected via PCA.

**Datasets and metrics.** We conduct experiments on MNIST, CIFAR, SVHN, and STL-10 and adopt standard metrics (cross-entropy, accuracy) together with a consistency measure that captures how stable predictions remain across composite networks built from different epochs.

**Point-wise consistency.** For a sample $(x, y)$ and checkpoints $T_1, T_2$, define

$$\text{Consistency}(x, y; T_1, T_2) = \frac{1}{|T_2 - T_1| + 1} \sum_{t=T_1 \wedge T_2}^{T_1 \vee T_2} \mathbb{I}\Big(f_{[d+1:n]}(\theta_{T_2}) \circ f_{[1:d]}(\theta_t)(x) = f(\theta_{t \vee T_2})(x)\Big)$$

where $\wedge, \vee$ denote min and max, respectively. This definition symmetrically covers both cases: for $T_1 < T_2$ it averages over shallow layers from $[T_1, T_2]$ with deep layers at $T_2$ (memory), and for $T_2 < T_1$ it averages over shallow layers from $[T_2, T_1]$ with deep layers at $T_2$ (transferability).

**Dataset-level consistency.** For memory experiments (with $t_1 < t_2$), we define dataset-level consistency as

$$\text{Consistency}_f(t_1, t_2) = \mathbb{P}_{(x,y) \sim \mu}\Big[f_{[d+1:n]}(\theta_{t_2})\Big(f_{[1:d]}(\theta_{t_1}, x)\Big) = f(\theta_{t_2})(x)\Big]$$

This measures, over the data distribution $\mu$, how often the composite model agrees with the reference network at the later epoch $t_2$. It serves as the main quantitative metric in our memory experiments.

## 4.2 MEMORY AND FORGETTING

**Memory**   In Experiment 1, composite networks retain high performance when the temporal gap between $t_1$ and $t_2$ is moderate. On CIFAR-10 (Fig. 2), once $t_1 \geq 150$, the composite network achieves near-zero cross-entropy and over 98% accuracy. Consistency values are tightly concentrated around 1.0, suggesting that for $t_1 \in [200, 300]$ most composite models correctly classify training samples. Experiments on test and OOD settings will be presented in Section 5.

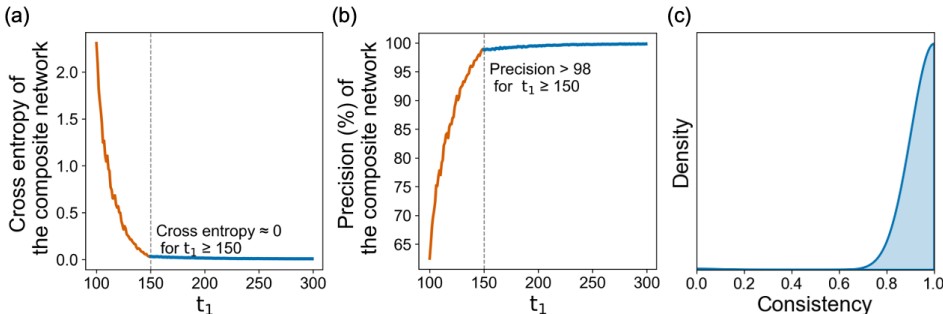

Figure 2: Performance of $f_{[d+1:n]}(\theta_{300}) \circ f_{[1:d]}(\theta_{t_1})$ on CIFAR-10, training set. (a) and (b) Cross-entropy and accuracy versus $t_1$. (c) Consistency across inputs for $t_1 \in [200, 300]$. Model: ResNet-20, $d$ at second basic block.

CAPABILITY STRENGTHENING OVER TRAINING.   As training proceeds, deep networks increasingly reuse earlier features. Consistency between composite and reference models is already high in early epochs, and the window above 0.8 gradually expands (Fig. 3), indicating classifiers become more robust to temporally varying shallow features. We explain in Appendix C for this phenomenon. This expanding consistency offers direct evidence that shallow layers act as structured augmenters, enabling deep classifiers to leverage temporal feature variations as implicit data augmentation.

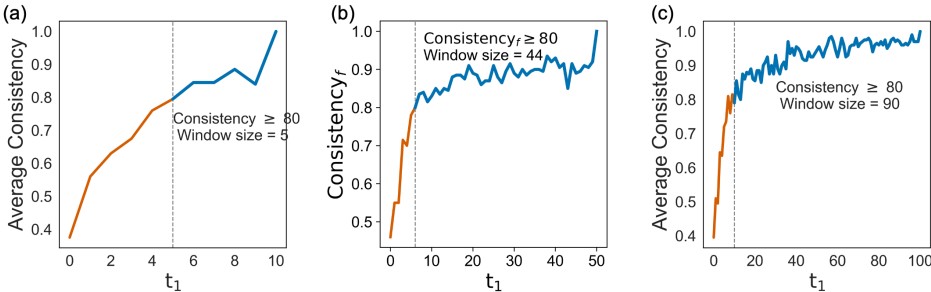

Figure 3: Consistency between composite networks $f_{[d+1:n]}(\theta_{t_2}) \circ f_{[1:d]}(\theta_{t_1})$ and $f(\theta_{t_1})$ during training. ResNet-20 on CIFAR-10. Consistency window expands over time, reaching > 0.8 threshold.

NOT A CONVERGENCE ARTIFACT.   Temporal consistency is not a trivial byproduct of convergence. As the gap $|t_2 - t_1|$ increases, composite accuracy gradually decays, contradicting the behavior expected under a stationary solution. Moreover, late-stage parameter drift (e.g., $\|\theta_{300} - \theta_{150}\|_2 / \|\theta_{300}\|_2 \approx 0.07$.) and non-negligible feature variation (Appendix Figure 11) indicate that the model continues to evolve even after loss and accuracy appear stable. This shows that the memory window reflects ongoing optimization dynamics rather than convergence or neural collapse.

**Forgetting**   As the temporal gap between $t_1$ and $t_2$ increases, composite performance deteriorates. On CIFAR-10 (Fig. 4), when $t_1$ decreases from 1000 to 300, cross-entropy rises and accuracy drops, indicating that later networks progressively forget features learned at earlier stages.

This phenomenon can not be explained with the neural tangent kernel (NTK) linearization (Jacot et al., 2018). Under this assumption, training reduces to convex optimization, where partial parameter

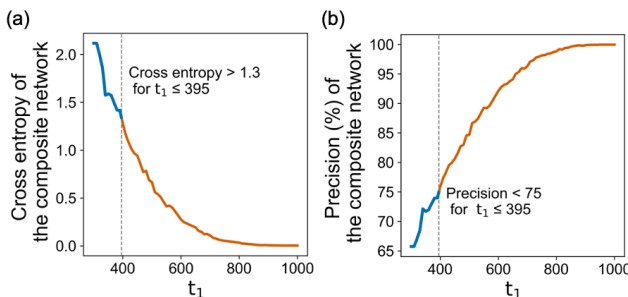

Figure 4: Performance of $f_{[d+1:n]}(\theta_{1000}) \circ f_{[1:d]}(\theta_{t_1})$ on CIFAR-10, training set. (a) and (b) Cross-entropy and accuracy drop as $t_1$ decreases, showing forgetting.

freezing would not reverse monotonic loss decrease. A detailed proof is provided in Appendix D, showing that the NTK regime prediction has no forgetting. The empirical forgetting we observe, therefore, reflects inherently nonlinear effects beyond NTK approximations.

### 4.3 TRANSFERABILIY

### 4.4 TRANSFERABILIY

In Experiment 2, we tested whether earlier classifiers can process features generated at the later stages. On CIFAR-10, when $t_2 \leq 500$, composite networks achieve near-zero cross-entropy and above 98% accuracy (Fig. 5). Consistency values are tightly concentrated around 1.0 for $t_2 \in [300, 500]$, showing that earlier classifiers remain predictive on the moderately shifted later features.

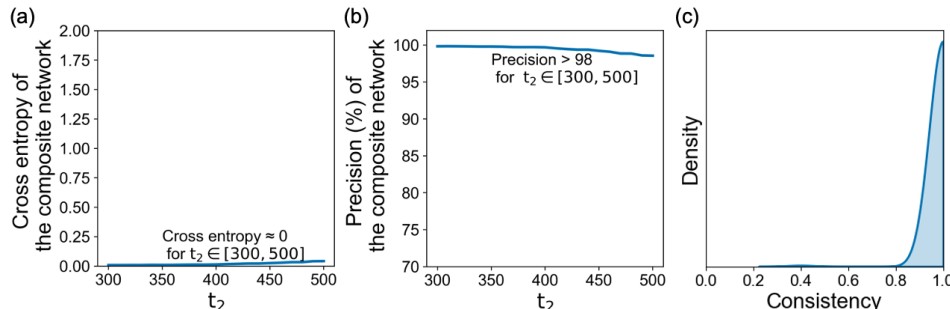

Figure 5: Transferability of $f_{[d+1:n]}(\theta_{300}) \circ f_{[1:d]}(\theta_{t_2})$ on CIFAR-10, training set. (a) and (b) Loss and accuracy versus $t_2$. (c) Consistency for $t_2 \in [300, 500]$. Model: ResNet-20, $d$ at second basic block.

### 4.5 FEATURE DYNAMICS INDUCES CLASSIFICATION REGION

In Experiment 3, we examined how evolving features related to the final decision boundaries. On MNIST (Fig. 6), features at epoch 200 form well-separated clusters, while earlier features (epochs 50–200) trace intermediate paths that fill the gaps between classes. Visualizing the classification regions at epoch 200 demonstrated distinct alignment with these historical trajectories, indicating that decision boundaries adapt to the regions explored by feature dynamics.

We further verify this phenomenon on CIFAR-10, where the latent space is high-dimensional. As detailed in Appendix 14, we project features onto principal components and add small perturbations. The results show that trajectories at later epochs remain stable along specific directions and are classified consistently by the final network. This robustness suggests that earlier feature paths leave a lasting imprint on the decision regions, reinforcing the view that feature dynamics guide the shaping of classification boundaries.

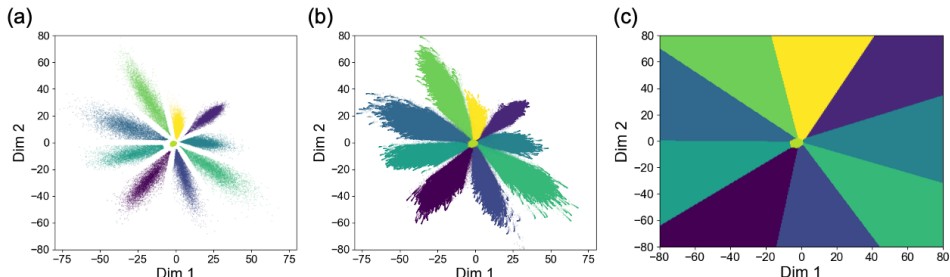

Figure 6: (a) and (b) Visualization of training features from a network at epoch 200 (a), and networks trained between epochs 51–200 (b), color-coded by class. (c) The classification regions of the epoch-200 network, colored by predicted class.

## 5 TEMPORAL AUGMENTATION EMERGES FROM GENERALIZABLE STRUCTURE

The structured temporal augmentations that arise during training are not mere memorization of samples. They generalize robustly to test and corrupted data, as long as training labels preserve semantic meaning. In contrast, with randomized labels, the model fails to exploit feature dynamics, and temporal consistency collapses.

**Extension to test and corrupted data.** We evaluated temporal consistency beyond training data on the CIFAR-10 test set and on CIFAR-10C with Gaussian blur (severity 5; (Hendrycks & Dietterich, 2019)). When trained with clean labels, consistency distributions remain sharply concentrated near 1.0 (Fig. 7(a,c) and Appendix Fig. 16), showing that training-induced temporal augmentations generalize to both unseen and corrupted inputs.

**Dependence on semantic labels.** In contrast, when 40% of training labels are randomized, the consistency distributions on both CIFAR-10 and CIFAR-10C collapse (Fig. 7(b,d) and Appendix Fig. 17): long tails emerge and mass shifts away from 1.0. This breakdown indicates that corrupted supervision prevents the model from reusing past features, confirming that temporal augmentation arises only when anchored in meaningful semantic structure. Statistical tests (Appendix E) further verify that the differences between clean and noisy-label training are significant.

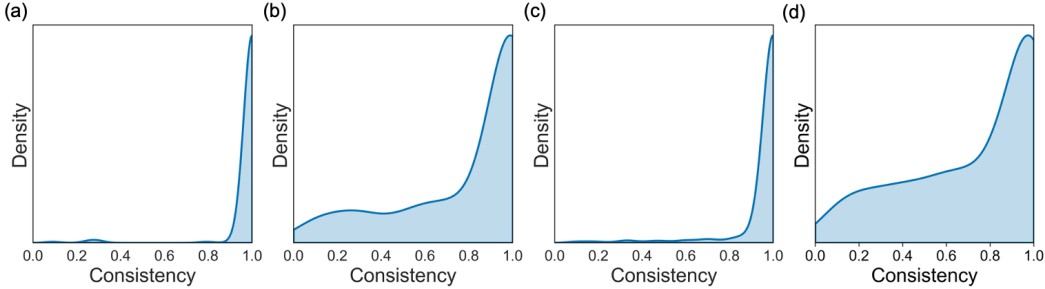

Figure 7: Consistency distributions across conditions. Temporal augmentation generalizes to the CIFAR-10 test set (a) and CIFAR-10C with Gaussian blur (c) when training labels are clean. With 40% random labels (b,d), consistency collapses, indicating that corrupted supervision prevents the model from exploiting past features.

**Depth-wide contribution.** We probe feature dynamics across depth by fixing shallow layers at later checkpoints and varying the treatment of deeper ones (Appendix A). Both reinitializing and retraining deeper layers under this setting lead to degraded accuracy and robustness, suggesting that the temporal evolution of shallow and deep representations jointly underpins generalization.

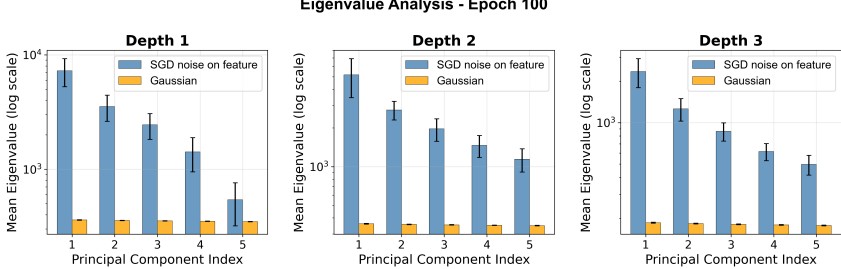

Figure 8: Top-5 eigenvalues of covariance for SGD-induced noise (blue) vs. isotropic Gaussian noise (orange) at epoch 100. See Appendix Fig. 15 for epochs 50 and 150.

# 6  STRUCTURED NATURE OF THE AUGMENTED FEATURE DISTRIBUTIONS

The temporal feature augmentations uncovered in previous sections are not arbitrary. Their distribution is shaped by structured variability injected during training, rather than by random isotropic noise.

**One-step perturbation protocol.**  To probe the mechanism, we performed controlled one-step updates with SGD. Fixing an input sample $x$ and a training checkpoint, we apply SGD updates with different mini-batches and measure the resulting feature changes $\Delta z$. Repeated perturbations yield a covariance matrix that characterizes the variability of the feature distribution around the current trajectory.

**Eigenvalue analysis.**  Eigenvalue analysis reveals that this covariance is far from isotropic. Instead of spreading variance evenly across all directions, the feature perturbations concentrate strongly along a few dominant axes. As shown in Fig. 8, the singular value spectrum of SGD-induced noise decays sharply, in contrast to the nearly flat spectrum produced by isotropic Gaussian perturbations of the same scale. Additional results at different epochs (Appendix Figure 15) confirm that this anisotropic pattern persists throughout training. To quantify the deviation from isotropy, we conducted a sphericity test on the normalized covariance. Across epochs and data points, the null hypothesis of isotropy is consistently rejected with extremely small $p$-values (often <0.001). This provides rigorous statistical evidence that SGD-induced noise forms a structured distribution rather than an unstructured cloud (Appendix E).

**Impacts**  These findings establish that temporal augmentations stem from low-dimensional, structured variability rather than random scatter. This predictability allows classifiers to exploit them, reinforcing their role as a reliable mechanism for generalization.

# 7  A CONCEPTUAL THEORY BRIDGE

Previous sections show that neural networks are robust to variations in their own learned features. Here, we outline a conceptual framework linking this robustness to generalization.

Let $f_{[1:n]}(\theta_t, \cdot)$ denote a neural network at training time $t$, with $f_{[1:d]}$ representing the shallower layers and $f_{[d+1:n]}$ the deeper layers. For a given input $x$ and reference time $t'$, we define the distribution of features produced by shallower networks within a time window $\Delta t$ as:

$$\omega_{x,t',\Delta t} := \text{distribution of } z_t(x) = f_{[1:d]}(\theta_t, x), \quad \text{where } t \sim \text{Uniform}\left([t' - \Delta t, t' + \Delta t] \cap \mathbb{Z}\right).$$

The corresponding augmented feature distribution for class $i$ over the training set is:

$$\Omega_{t',\Delta t}^{(i)} := \int_{\mathcal{X}} \omega_{x,t',\Delta t} \, d\bar{\mu}_i(x),$$

where $\bar{\mu}_i$ is the empirical distribution for class $i$ over the training data. We say the network exhibits empirical-level memory and transferability at $(\delta, \Delta t)$ and epoch $t'$ if the classifier gives consistent predictions on most feature perturbations (up to error $\delta$), relative to those from time $t'$.

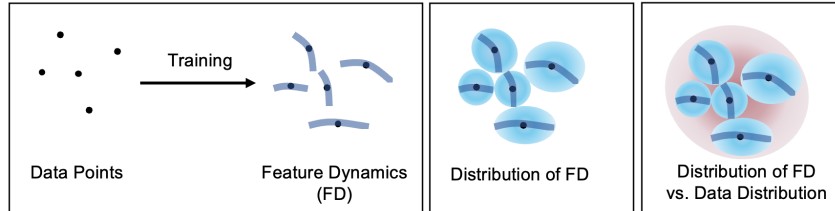

Figure 9: Conceptual illustration of the temporal augmentation framework. (Left & Middle) Shallow network $f_{[1:d]}$ generates temporally varying features (blue trajectories), deep classifier $f_{[d+1:n]}$ processes variations (blue spheres). (Right) Discrepancy between temporally augmented features (blue) and test distribution (red).

**Theorem 1** (Informal). *If empirical-level memory holds at $(\delta, \Delta t)$, then the generalization gap for label $i$ satisfies:*

$$\left| \int_{\mathcal{X}} \mathbb{I}(f(\theta_{t'}, x) \neq i) \, d\mu_i - \int_{\mathcal{X}} \mathbb{I}(f(\theta_{t'}, x) \neq i) \, d\bar{\mu}_i \right| \leq TV\left( \Omega_{t',\Delta t}^{(i)} \,\middle\|\, f_{[1:d](\theta_{t'})\#}(\mu_i) \right) + \delta,$$

*where $\mu_i$ is the true distribution of class $i$, $TV(\cdot \| \cdot)$ denotes total variation distance, $f_{\#}(\mu_i)$ is the pushforward measure (Appendix A, D) and $\Omega_{t',\Delta t}^{(i)}$ is the augmented feature distribution for class $i$.*

See Appendix G for the proof. Figure 9 illustrates how temporal feature variations serve as implicit augmentation and clarifies the TV term.

Unlike traditional i.i.d.-based generalization theories, our conceptual framework incorporates training-induced data augmentation and assesses generalization in latent space. This perspective may offer insights into the temporal dynamics of feature representations. Empirically, small prediction thresholds ($\delta$) often correspond to large time windows ($\Delta t$), suggesting a broader generalizable region than conventional analyses imply. However, the framework remains theoretical due to two intractable components: the feature distribution $\omega_{x,t',\Delta t}$ and total variation distance. Choosing appropriate values for $\delta$ and latent depth $d$ is also challenging. Developing practical estimators or proxies for these terms represents a key direction for future work.

## 8 CONCLUSION

Generalization in deep learning remains only partially understood. In this work, we focus on the dynamics of features themselves. Our analysis highlights a simple but powerful observation: temporal consistency—the stability of predictions when mixing shallow and deep features across training—functions as a structured form of augmentation.

Through experiments on clean, corrupted, and noisy-label settings, as well as statistical evidence on the anisotropy of SGD noise, we showed that this consistency is not an artifact of memorization but a property that actively supports robustness.

We conclude by offering a perspective: temporal consistency provides a bridge between feature dynamics and generalization. While still conceptual, this framing opens avenues for future research toward tractable surrogates and profound insights, potentially connecting with divergences such as MMD or Wasserstein distances.

## 9 DISCLOSURE

We used large language models (e.g., ChatGPT) solely for language polishing and grammar improvement. In addition, we used an LLM-based coding assistant (Cursor) to aid in code editing and debugging. No experimental design, or analysis was generated by LLMs.

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

APPENDICES

# A    FEATURE DYNAMICS AND NOTATION

## A    PARAMETER DYNAMICS

Parameter dynamics refers to the evolution of parameters driven by optimization algorithms within the corresponding parameter space. A commonly used class of algorithms in this context is gradient-based optimization. Specifically, consider a training set $\{(x_i, y_i) \in \mathcal{X} \times \mathcal{Y} \mid i = 1, 2, \ldots, N\}$, where the pairs $(x_i, y_i)$ are independently and identically distributed (i.i.d.) samples, and let $l : \mathcal{X} \times \mathcal{Y} \to \mathbb{R}$ be a loss function. The empirical loss is then defined as $L = \frac{1}{N} \sum_{i=1}^{N} l(x_i, y_i)$.

The gradient descent method updates the parameters according to:

$$\theta_{t+1} = \theta_t - \eta \cdot \nabla L$$

where $\eta$ is the step size. Due to the size of training set, people often use stochastic gradient descent (SGD) to update the parameters, which approximate $L$ by $L'$, a batch of randomly sampled data $\mathcal{B}$: $L' = \frac{1}{|\mathcal{B}|} \sum_{i \in \mathcal{B}} l(x_i, y_i)$. As a result, optimization can be viewed as continuously adding noise $\nabla L' - \nabla L$ into the original optimization process. In this perspective, the gradient method is equivalent to :

$$\theta_{t+1} = \theta_t - \eta \cdot \nabla L + \eta \cdot \text{noise}$$

**Continuous form of parameter dynamics.**    The negative gradient flow is usually concerned as the continuous form of the gradient descent method since its Euler discretization corresponds to gradient descent:

$$\frac{d\theta_t}{dt} = -\nabla L$$

Similarly, people formulate SGD by the stochastic differential equation Chen et al. (2022); Chaudhari & Soatto (2018):

$$d\theta_t = -\nabla L \, dt + \sqrt{\Sigma} \, dB_t$$

where $\Sigma$ denotes the noise covariance matrix, and $B_t$ is a high-dimensional Brownian motion vector with the same dimension as $\theta$. Through this formula, we gain an intuitive understanding of how the optimization algorithm injects noise into the neural network.

## B    FEATURE DYNAMICS (DURING TRAINING)

The changes in parameters during optimization imply that the mappings of each layer in the network are evolving, and the features of each hidden layer are continuously changing. This phenomenon is referred to as feature dynamics. Specifically, given the parameters of the first layer $\theta^1$, denote the corresponding network layer by $f_1(\theta^1, \cdot)$. The hidden feature of data $x$ is given by $f_1(\theta_t^1, x)$. The dynamics of hidden feature in the first layer during training is given by:

$$z_t^1 = f_1(\theta_t^1, x), \quad z_{t+1}^1 = z_t^1 + \left( f_1(\theta_{t+1}^1, x) - f_1(\theta_t^1, x) \right)$$

By $z_t^{i+1} = f(\theta_i, z_t^i)$, it is easy to derive the dynamics of hidden feature in any layer.

**Continuous form of feature dynamics**    Through a Taylor expansion, we can directly observe how feature dynamics evolve in continuous time for a fixed $x$:

$$f_{1,k}(\theta_t^1 + \Delta\theta, x) = f_{1,k}(\theta_t^1, x) + \nabla_\theta f_{1,k}(\theta_t^1, x) \cdot \Delta\theta + \frac{1}{2}\Delta\theta^T \cdot \nabla_\theta^2 f_{1,k}(\theta_t^1, x) \cdot \Delta\theta + \text{Higher order}$$

where $f_1 = [f_{1,1}, \cdots, f_{1,w_1}]$, $w_1$ represents the width of the first layer. Using the fundamental concepts of (stochastic) calculus and continuous form of SGD, the above equation can be further

transformed:

$$dz_{t,k}^1 = df_{1,k}(\theta_t^1, x) \tag{1}$$

$$= \frac{df_{1,k}}{d\theta}(\theta_t^1, x)d\theta + \frac{1}{2}\text{tr}\left(\nabla_\theta^2 f_{1,k}(\theta_t^1, x)[d\theta, d\theta]\right) \tag{2}$$

$$= \underbrace{\left(-\frac{df_{1,k}}{d\theta}\nabla L + \frac{1}{2}\text{tr}\left(\nabla_\theta^2 f_{1,k}(\theta_t^1, x)\Sigma\right)\right)dt}_{\text{deterministic part}} + \underbrace{\frac{df_{1,k}}{d\theta}\sqrt{\Sigma}\,dB_t}_{\text{random part}} \tag{3}$$

where $z_t^1 = [z_{t,1}^1, \cdots, z_{t,w_1}^1] = f_1(\theta_t^1, x)$. By $z_t^{i+1} = f_{i+1}(\theta_t^i, z_t^i)$ and Ito formula, one can derive the continuous form of feature dynamics in any layer. Through these formula, we establish a connection between parameter dynamics and feature dynamics, directly illustrating how the optimization algorithm injects noise into the hidden layer features of the neural network. It is important to emphasize that the noise in the parameters propagates to the hidden layer features, causing the feature of a single data point in the hidden layer to no longer be an isolated point, but rather a distribution. This may enhance the effectiveness of individual data points.

## C  CLASSIFICATION REGION

Given a classifier $f : \mathbb{R}^n \to \mathcal{Y}$, the classification region refers to the set of all elements in the domain of $f$ that are assigned to a particular label by $f$. For example, given a bi-classifier $f : \mathbb{R}^n \to \{-1, 1\}$, it's classification regions are $A_1 = f^{-1}(1)$ and $A_{-1} = f^{-1}(-1)$, where $f^{-1}(y) = \{x, f(x) = y\}$.

Given a classification deep neural network $f(\theta_t, \cdot) = f_{[1:n]}(\theta_t, \cdot) := f_n(\theta_t^n, \cdot) \circ \cdots \circ f_2(\theta_t^2, \cdot) \circ f_1(\theta_t^1, \cdot)$, where $f_i(\theta_t^i, \cdot)$ represents the $i$-th layer at epoch $t$, $n$ is the depth of network, the symbol $\circ$ denotes the composition of functions. The classification region of network $f_{[1:n]}$ of the $d$-th hidden feature space is defined by the classification region of $f_{[d+1:n]}(\theta_t, \cdot) := f_n(\theta_t^n, \cdot) \circ \cdots \circ f_{d+1}(\theta_t^{d+1}, \cdot)$.

Intuitively, the classification region describes certain robustness of the classification function. In fact, this concept is closely related to generalization: if the support of the true data distribution for a particular class entirely lies within the classification region corresponding to the label of the classification function, then the classifier's generalization error for that class's data is zero.

## D  INDUCED MEASURE

Given two measurable spaces $\mathcal{X}$ and $\mathcal{Y}$, along with a measure $\mu$ on $\mathcal{X}$, if there exists a function $f : \mathcal{X} \to \mathcal{Y}$, then $f$ induces a measure $f_\#(\mu)$ on $Y$, which is given by:

$$f_\#(\mu)(A) = \mu\left(f^{-1}(A)\right)$$

where $A$ is a measurable set of $\mathcal{Y}$ and $f^{-1}(A) = \{x, f(x) \in A\}$. $f_\#(\mu)$ is also referred to as the pushforward measure or the image measure of $\mu$ under $f$. It describes how the measure $\mu$ on $\mathcal{X}$ is transferred to $\mathcal{Y}$ via the function $f$.

Denote the empirical distribution (distribution of training set) as $\bar{\mu}$, the true data distribution as $\mu$, and the corresponding distribution of samples with label $i$ as $\bar{\mu}_i$ and $\mu_i$. Then, in the hidden space of the $d$-th layer of the neural network $f_{[1:n]}$, their corresponding induced distributions are given by $f_{[1:d]\#}(\bar{\mu})$, $f_{[1:d]\#}(\mu)$, $f_{[1:d]\#}(\bar{\mu}_i)$ and $f_{[1:d]\#}(\mu_i)$. As previously noted (3), noise is continuously injected during training, transforming the induced empirical distributions from a combination of delta functions (where probability mass is concentrated at single points) into outcomes of stochastic processes. These noise-injected distributions have broader support than delta combinations, potentially aiding in the understanding of robustness and generalization.

## B Experiments Figure

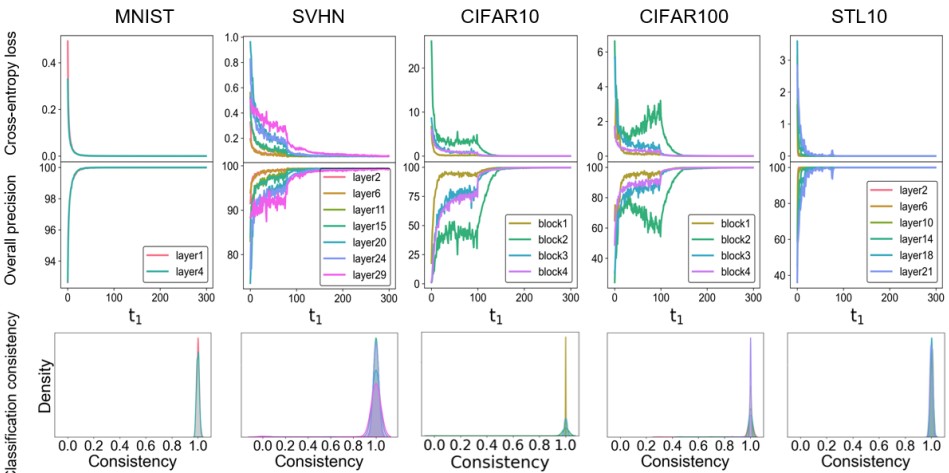

Figure 10: Memory: The curves in the first two lines describe how the Cross Entropy Loss and classification accuracy of the combined network change as $t_1$ varies when $t_2 = 300$. The different numbers of layers/blocks represent the number of layers $d$ in the network with epoch $t_1$. The last row estimates the density of Consistency in the empirical distribution through sampling and kernel density estimation.

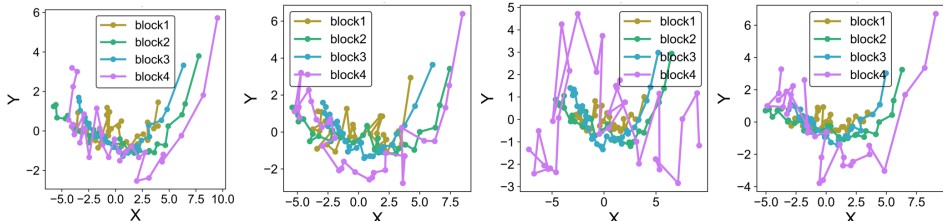

Figure 11: Memory: Each figure presents the features of randomly selected data extracted from shallower networks trained for different numbers of epochs. Each point in the figure represents a feature extracted by a specific shallower network, with different colors indicating the corresponding layer of the shallower network. Features were extracted every five epochs, and we visualized those at epochs 150–300.

**Memory:**

**Forgetting:** For CIFAR-10, CIFAR-100, and SVHN, when $t_1$ ranges from 300 to 1000, a decrease in $t_1$ leads to an increase in the cross-entropy loss of the constructed network and a corresponding decrease in classification accuracy. In contrast, for the MNIST and STL-10 datasets, accuracy remains largely unchanged as $t_1$ varies. This stability may be attributed to the relative simplicity of these tasks, allowing the network to quickly converge to a local minimum.

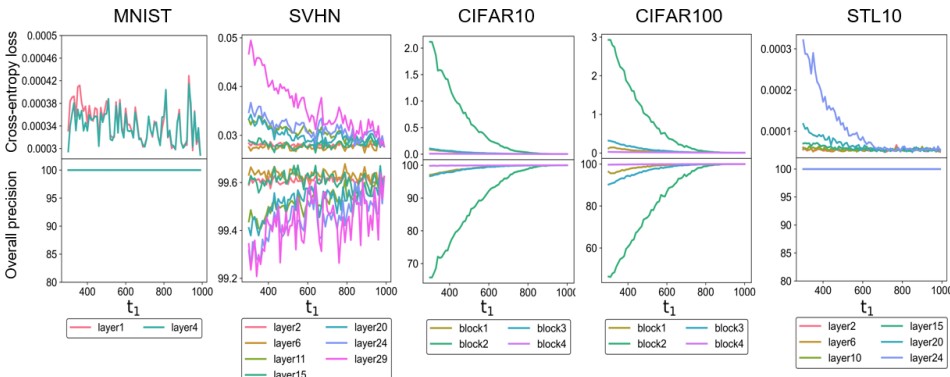

Figure 12: Forgetting. The curves in the first two lines describe how the Cross Entropy Loss and classification accuracy of the constructed network change as $t_1$ varies when $t_2 = 1000$. The different numbers of layers/blocks represent the number of layers $d$ in the network with epoch $t_1$. This results in a relatively significant decline in network performance.

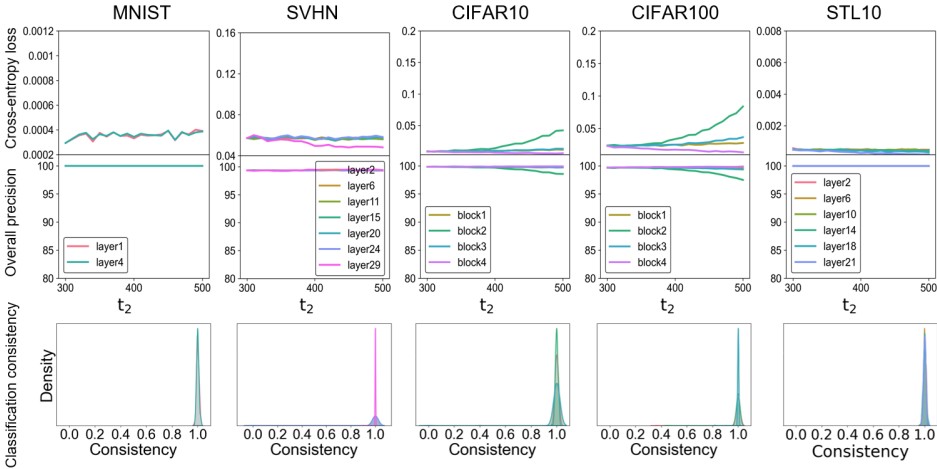

Figure 13: Transferability. The curves in the first two lines describe how the Cross Entropy Loss and classification accuracy of the constructed network change as $t_2$ varies when $t_1 = 300$. The different numbers of layers/blocks represent the number of layers $d$ in the network with epoch $t_2$. As shown in the figure, when $t_2$ ranges from 300 to 500, the constructed network's Cross Entropy Loss approaches 0, and the classification accuracy remains high. The last row estimates the density of Consistency in the empirical distribution through sampling and kernel density estimation.

**Transferability:**

**Induction** For the CIFAR-10 dataset, the network's latent space is high-dimensional. To analyze feature dynamics, we applied PCA to identify the two principal directions and examined the stability of feature trajectories along these directions under perturbations. Specifically, we added uniform noise of approximately unit magnitude (as shown in Figure 14) to features at epochs 250–300, along the two principal components, and evaluated their classification using the network at epoch 300. This experiment reveals the robustness of feature dynamics in specific directions. The observed stability in certain directions suggests that, in high-dimensional space, feature dynamics are closely related to classification regions. This implies that features extracted in earlier epochs retain partial robustness and that classification boundaries are significantly influenced by the evolution of feature dynamics.

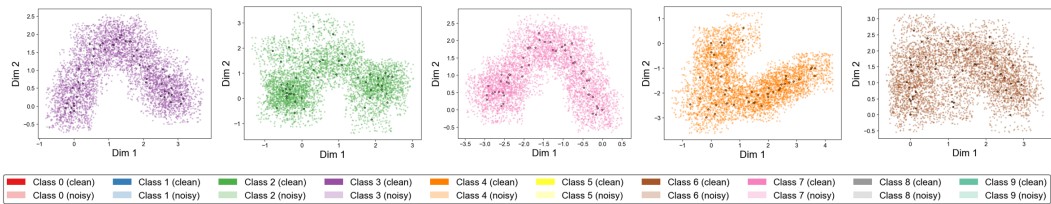

Figure 14: Induction in CIFAR10. We present the feature dynamics of five randomly selected samples, along with their corresponding classification results after noise perturbation. Darker points represent the projections of feature dynamics (at epochs 250–300) onto a two-dimensional plane, while lighter points surrounding each dark dot indicate the network's predictions after noise is applied. Each point is colored according to the label assigned by the deeper layers of the network at epoch 300.

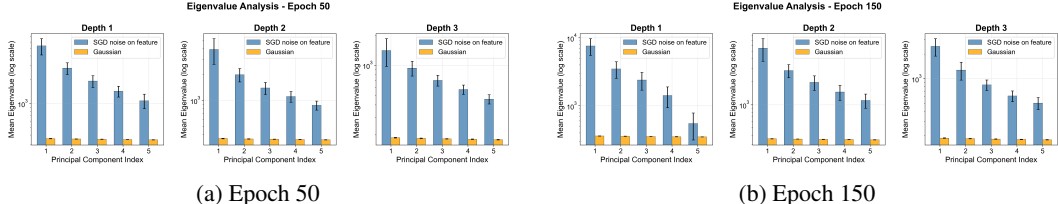

(a) Epoch 50                                    (b) Epoch 150

Figure 15: Top-5 eigenvalue spectra of SGD-induced noise (blue) vs. isotropic Gaussian noise (orange) at additional training epochs. The anisotropic pattern is consistent with the 100-epoch result in the main text.

**Top-5 eigenvalues of SGD-induced noise**

**CIFAR-10C Robustness Analysis** To demonstrate the robustness of temporal augmentation under distribution shift, we evaluated ResNet-20 networks (trained on CIFAR-10) on CIFAR-10C with severity=5 corruption. Figure 16 shows the consistency distributions across different noise types for epochs 200-300. The high concentration of consistency values near 1.0 across all corruption types indicates that temporal augmentation remains effective even under severe distribution shift, demonstrating the structured nature of the augmentation mechanism.

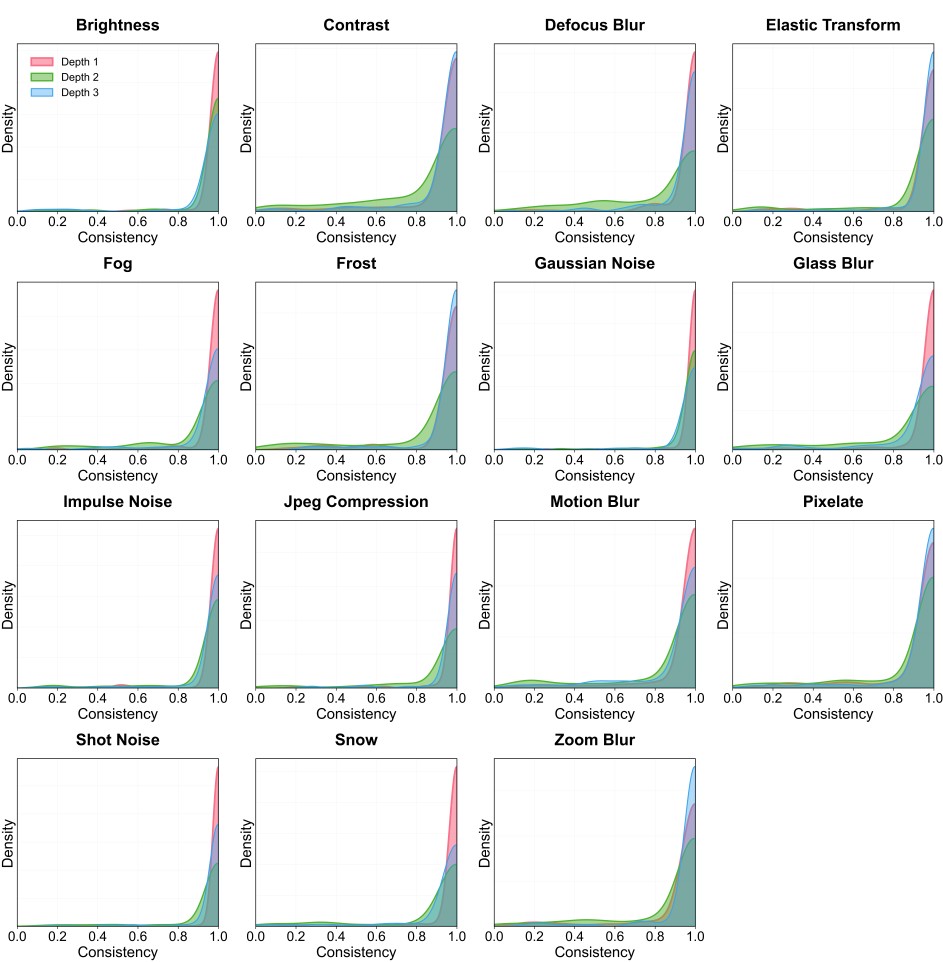

Figure 16: Consistency distributions on CIFAR-10C with severity=5 corruption for ResNet-20 networks, trained on clean dataset (epochs 200-300). The high concentration of consistency values near 1.0 across all corruption types (Gaussian noise, shot noise, impulse noise, defocus blur, glass blur, motion blur, zoom blur, snow, frost, fog, brightness, contrast, elastic transform, pixelate, JPEG compression) demonstrates that temporal augmentation remains effective under severe distribution shift, indicating the structured nature of the augmentation mechanism.

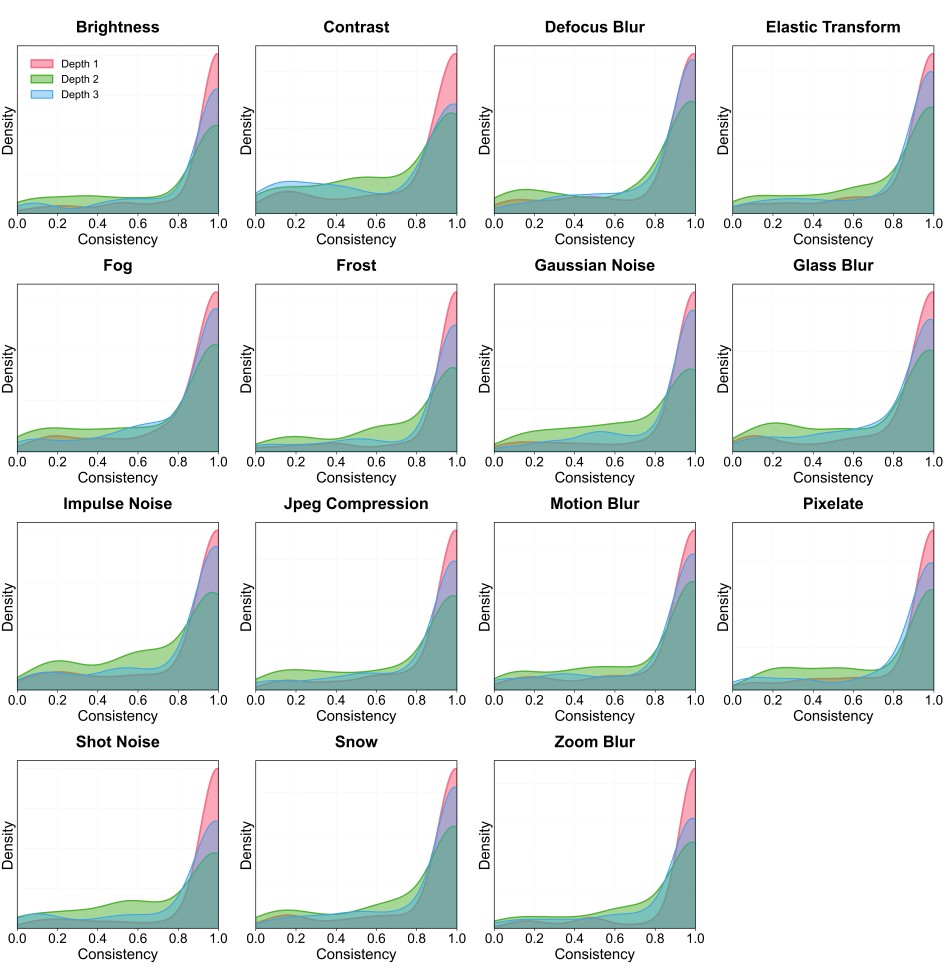

Figure 17: Consistency distributions on CIFAR-10C with severity=5 corruption for ResNet-20 networks, trained on 40% noised label (epochs 200-300). The long tail demonstrates that corrupted supervision prevents the model from reusing past features, confirming that temporal augmentation arises only when anchored in meaningful semantic structure.

## C   ON THE MEMORY PHENOMENON IN TRAINING

Consider a neural network decomposed as

$$f(x) = f \circ g(x),$$

where $g$ denotes the shallow part and $f$ the deep part. Let

$$z_t = g_t(x)$$

denote the hidden feature at training step $t$. As training goes, the overall mapping satisfies

$$f_t(g_t(x)) \to C,$$

where $C$ is nearly constant on the training set. Differentiating with respect to training time $t$ yields

$$\frac{\partial f}{\partial z}\frac{dz}{dt} + \frac{\partial f}{\partial \theta}\frac{d\theta}{dt} \to 0.$$

The parameter sensitivity $\frac{\partial f}{\partial \theta}$ is getting small, so the dominant balance is

$$\frac{\partial f}{\partial z}\frac{dz}{dt} \to 0.$$

This balance explains why deep classifiers remain predictive on features generated at earlier epochs: features drift in directions that are locally "flat" for the classifier. This partly explains the memory phemenon.

However, this explanation is specific to the training set. On test data or corrupted inputs, the shallow feature trajectories differ, and the balance above no longer holds globally. Thus, while the gradient balance clarifies why training samples exhibit memory, generalization beyond training requires the structured augmentation view developed in the main text.

# D THE NTK ASSUMPTION AND FORGETTING

**Definition 1** (The Neural Tangent Kernel (NTK) assumption (Linear assumption)). *NTK hypothesis posits that when training deep neural networks, if the network parameters are properly initialized and the network is sufficiently large, the behavior of the network can be approximated through linearization. In other words, the network's output, with respect to changes in the parameters, is approximately linear:*

$$f(\theta_t, x) \approx f(\theta_0, x) + \nabla_\theta f(\theta_0, x)(\theta_t - \theta_0)$$

*where $f(\theta_t, x)$ represents the function of the neural network at optimization epoch t. For simplicity, we assume that:*

$$f(\theta_t, x) = f(\theta_0, x) + \nabla_\theta f(\theta_0, x)(\theta_t - \theta_0)$$

*which is a common setting in many theoretical analysis.*

**Lemma 1** (Logarithm of "Soft-max" is "concave"). *The $-\log s_j(\cdot)$ is convex, where the "Soft-max" function $s_j$ is defined as:*

$$s_j : \mathbb{R}^n \to \mathbb{R}$$

$$s_j(x) = \frac{\exp(x_j)}{\sum_i \exp(x_i)}$$

*where k is a fixed index.*

**Proof**:

1. It's obvious that:

$$\sum_j s_j = 1$$

2. The partial derivative of $s_j$ is:

$$\frac{\partial}{\partial x_j} s_j = \frac{\exp(x_j)}{\sum_i \exp(x_i)} - \frac{\exp(x_j)^2}{\left(\sum_i \exp(x_i)\right)^2} = s_j - s_j^2$$

$$\frac{\partial}{\partial x_k} s_j = \frac{-\exp(x_k)\exp(x_j)}{\left(\sum_i \exp(x_i)\right)^2} = -s_k s_j$$

where $k \neq j$.

3. (By 2,) We know that the first order derivative of $-\log s_j$ is:

$$\frac{\partial}{\partial x_j}(-\log s_j) = \frac{-s_j + s_j^2}{s_j} = -1 + s_j$$

$$\frac{\partial}{\partial x_k}(-\log s_j) = \frac{s_j s_k}{s_j} = s_k$$

where $k \neq j$. The second order derivative of $-\log s_j$ is:

$$\frac{\partial^2}{\partial x_j^2}(-\log s_j) = s_j - s_j^2 > 0$$

$$\frac{\partial^2}{\partial x_j x_k}(-\log s_j) = -s_k s_j < 0$$

$$\frac{\partial^2}{\partial x_k^2}(-\log s_j) = s_k - s_k^2 > 0$$

$$\frac{\partial^2}{\partial x_k x_l}(-\log s_j) = -s_k s_l < 0$$

where $k \neq j$ and $k \neq l$.

4. (By 1 and 3,) Hessian of $-\log s_j$ is diagonally dominant:

$$\sum_{k, k \neq j} \frac{\partial^2}{\partial x_j x_k}(-\log s_j) = \sum_{k, k \neq j} -s_k s_j = s_j^2 - s_j = \frac{\partial^2}{\partial x_j^2} \log s_j$$

$$\sum_{l, l \neq k} \frac{\partial^2}{\partial x_k x_l}(-\log s_j) = \sum_{l, l \neq k} -s_k s_l = s_k^2 - s_k = \frac{\partial^2}{\partial x_k^2} \log s_j$$

As a result, the $-\log s_j$ is a convex function and the gradient flow can converge to the global minima.

$\square$

By linear assumption and the cross-entropy loss, we know that the loss function can be written as:

$$\text{Loss}(f(\theta_t, \cdot)) = -\sum_i \log s_{y_i}\big(f(\theta_0, x_i) + \nabla_\theta f(\theta_0, x_i)(\theta_t - \theta_0)\big)$$

This is a convex function with respect to parameter $\theta$. By gradient descent (gradient flow), the value of the loss function decrease monotonicly, even when some of the parameters are frozen. As a result:

$$\text{Loss}(f_{[d:n]}(\theta_{t_2}, \cdot) \circ f_{[1:d]}(\theta_{t_1}, \cdot)) \leq \text{Loss}(f_{[d:n]}(\theta_{t_1}, \cdot) \circ f_{[1:d]}(\theta_{t_1}, \cdot)) = \text{Loss}(f(\theta_{t_1}, \cdot))$$

as long as $t_1 < t_2$. However, the Forgetting phenomenon shows that $\text{Loss}(f_{[d:n]}(\theta_{t_2}, \cdot) \circ f_{[1:d]}(\theta_{t_1}, \cdot))$ getting bigger than $\text{Loss}(f(\theta_{t_1}, \cdot))$, which is contradict with the theoretical analysis. Hence the NTK assumption cannot explain the phenomenon.

# E  HYPOTHESIS TESTING

**Welch's t-test Analysis**  To confirm that the differences in consistency distributions between clean and noisy-label training are statistically significant, we conduct Welch's t-test. Given two sets of samples with means $\bar{x}_1, \bar{x}_2$, variances $s_1^2, s_2^2$, and sizes $n_1, n_2$, the test statistic is

$$t = \frac{\bar{x}_1 - \bar{x}_2}{\sqrt{s_1^2/n_1 + s_2^2/n_2}},$$

with degrees of freedom estimated using the Welch–Satterthwaite equation

$$\nu \approx \frac{(s_1^2/n_1 + s_2^2/n_2)^2}{\frac{(s_1^2/n_1)^2}{n_1-1} + \frac{(s_2^2/n_2)^2}{n_2-1}}.$$

Table 1 summarizes the $p$-values for representative comparisons. In all cases, $p < 0.05$, confirming that the observed differences are statistically significant.

Table 1: Welch's t-test results comparing consistency distributions under different label conditions.

| Comparison(Clean vs. 40% noisy labels) | $p$-value |
|---|---|
| Depth 1 | $5.46244 \times 10^{-06}$ |
| Depth 2 | $2.02578 \times 10^{-14}$ |
| Depth 3 | $5.46244 \times 10^{-06}$ |

**Isotropy Test Methodology**  We test whether SGD-induced noise is isotropic. Given feature perturbations $\Delta z^{(b)}$ from repeated one-step SGD, we estimate the sample covariance $S \in \mathbb{R}^{d \times d}$. To remove overall scale, we trace-normalize

$$\tau = \frac{\mathrm{tr}(S)}{d}, \qquad \widetilde{S} = \frac{S}{\tau}.$$

The null hypothesis is isotropy,

$$H_0 : \Sigma = \sigma^2 I_d,$$

and our test statistic is the Frobenius deviation from identity:

$$T = \tfrac{1}{d}\|\widetilde{S} - I_d\|_F^2.$$

We calibrate $T$ via a parametric bootstrap: generate $B$ Gaussian matrices with the same $(N, d)$, compute $T^{(b)}$ for each, and report the right-tail $p$-value

$$p = \frac{1 + \sum_{b=1}^{B} \mathbf{1}\{T^{(b)} \geq T_{\mathrm{obs}}\}}{B + 1}.$$

Across epochs and data points, observed $p$-values are consistently very small (typically $< 0.001$), rejecting the isotropic null and confirming that SGD-induced noise is anisotropic.

# F  SUPPORTING EVIDENCE

## A  REINITIALIZING DEEP LAYERS

We further probe the role of feature dynamics across depth by selectively freezing shallow layers of a well-trained ResNet-20 and reinitializing deeper layers before retraining on CIFAR-10.

**Clean test accuracy.**  When only the first basic block is retained, the final test accuracy decreases from $92.0\%$ to $91.2\%$. Retaining the first two basic blocks results in $91.5\%$ accuracy. Although the degradation is modest, these results indicate that preserving shallow representations alone is insufficient to fully recover the original performance.

**Robustness under corruption.**  We also evaluate robustness on CIFAR-10C. Following our implementation, the mean corruption error (mCE) is defined as the unnormalized average error across all 15 corruption types and 5 severities:

$$\mathrm{mCE} = \frac{1}{15 \times 5} \sum_{\mathrm{corr},s} (1 - \mathrm{acc}_{\mathrm{corr},s}).$$

For the baseline model, mCE is $0.316$. After reinitializing deeper layers, mCE increases to $0.326$ (first block retained) and $0.329$ (first two blocks retained). This corresponds to a $3$–$4\%$ accuracy drop under corruptions, suggesting that temporal consistency throughout the depth of the network plays an important role in robustness, i.e., in a broader sense of generalization.

**Summary.**  These results provide supporting evidence that while shallow features capture transferable structure, maintaining the feature dynamics contributes both to clean accuracy and to robustness under distribution shift.

## B  FREEZING SHALLOW LAYERS

At epoch 150, we freeze the shallow network up to depth d and continue training the deeper layers using SGD. This setup isolates the effect of shallow feature drift by preventing further updates in early layers. The results show that models with frozen shallow networks (Fix Depth1/2/3) achieve slightly lower and less stable test accuracy compared to the fully trainable baseline (SGD) (Figure 18). This indicates that allowing shallow layers to continue evolving provides a positive contribution to generalization: feature drift in early layers acts as a form of structured augmentation that benefits the deep classifier.

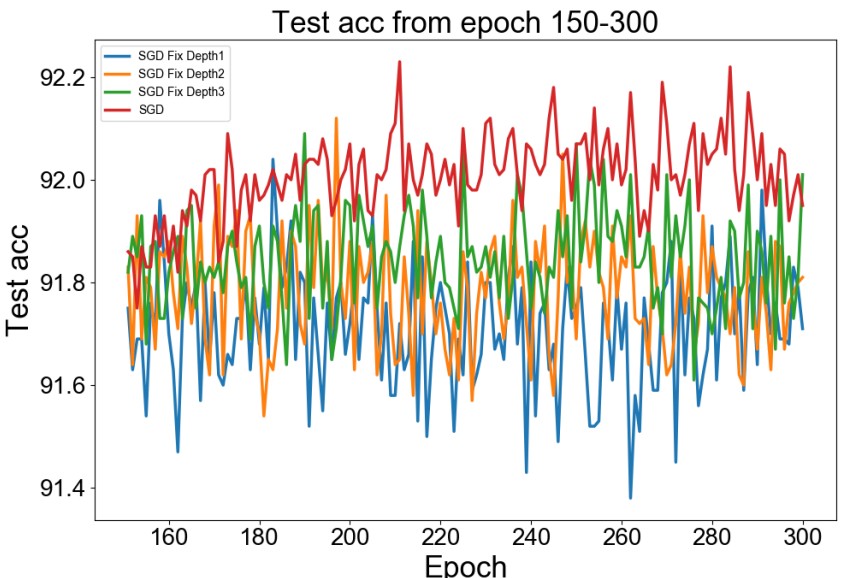

Figure 18: Test accuracy with and without shallow network freezing (epochs 150–300).

## G MEMORY AND GENERALIZATION

### A MATHEMATICAL FORMULATION OF OBSERVATIONS

Let $f_{[1:n]}(\theta, \cdot)$ denote a trained neural network. Given input data $x$, depth $d$, and training time $t$, the network maps $x$ to $z_t(x)$. Owing to the stochastic nature of parameter dynamics, the quantity $\{z_t(x)\}$ constitutes a random variable. Assume the training epoch $t'$ is given, $t$ follows a uniform distribution, and let the probability measure of $z_t(x)$, where $t \in [t' - \Delta t, t' + \Delta t] \cap \mathbb{Z}$, be denoted by $\omega_{x,t',\Delta t}(z)$. We formalize the memory phenomenon as follows:

**Definition 2** (Memory and Transferability). *We define Memory and Transferability (abbreviated as memory) at two levels of granularity:*

- *(Particle-level) Given a data $x$, the family of networks $\{f(\theta_t, \cdot)\}_{t \geq 0}$ is said to exhibit particle-level memory at $(\delta, \Delta t)$ and epoch $t'$ for $x$, if the following condition holds:*

$$\int_{\mathcal{Z}} \mathbb{I}\left(f_{[d+1:n]}(\theta_{t'}, z) - f_{[d+1:n]}(\theta_{t'}, z_{t'}(x)) = 0\right) d\omega_{x,t',\Delta t}(z) \geq 1 - \delta$$

  *where $\mathbb{I}(x = 0) = 0$ if $x \neq 0$. Otherwise, $\mathbb{I}(x = 0) = 1$.*

- *(Empirical-level) Denote the empirical distribution of data as $\bar{\mu}(x)$. the family of networks $\{f(\theta_t, \cdot)\}_{t \geq 0}$ is said to exhibit empirical-level memory at $(\delta, \Delta t)$ and epoch $t'$, if the following condition holds:*

$$\int_{\mathcal{Z} \times \mathcal{X}} \mathbb{I}\left(f_{[d+1:n]}(\theta_{t'}, z) - f_{[d+1:n]}(\theta_{t'}, z_{t'}(x))\right) d\omega_{x,t',\Delta t}(z) d\bar{\mu}(x) \geq 1 - \delta$$

  *where $f_{[d+1:n]}$ represent the layers after depth $d$.*

The particle-level memory phenomenon refers to a fixed sample being consistently processed by updated networks using features extracted by shallower networks. In contrast, empirical-level memory captures this behavior across samples drawn from the data distribution. Our observations indicate that, as training progresses, neural networks tend to exhibit empirical-level memory, with most samples showing particle-level memory. Moreover, a relatively small $\delta$ often corresponds to a relatively large $\Delta t$, offering insights into generalization behavior.

Clearly, if all samples exhibit particle-level memory, empirical-level memory naturally follows. However, the converse does not necessarily hold. We now formalize the relationship between these two levels of memory.

**Theorem 2.** *(1) Given a family of networks $\{f(\theta_t, \cdot)\}_{t \geq 0}$, if particle-level memory holds at $(\delta, \Delta t)$ and epoch $t'$ for every $x$ in the data set, then empirical-level memory at $(\delta, \Delta t)$ will occur.*

*(2) If empirical-level memory holds at $(\delta, \Delta t)$ and epoch $t'$, then more than $1 - \frac{1}{q}$ of the data in the empirical distribution exhibit a particle-level memory property at $(q\delta, \Delta t)$ and epoch $t'$, for any $q > 1$.*

**Proof:**

(1) By particle-level memory, we have:

$$\int_{\mathcal{Z}} \mathbb{I}\left(f_{[d+1:n]}(\theta_{t'}, z) - f_{[d+1:n]}(\theta_{t'}, z_{t'}(x)) = 0\right) d\omega_{x,t',\Delta t}(z) \geq 1 - \delta$$

Therefore,

$$\int_{\mathcal{Z} \times \mathcal{X}} \mathbb{I}\left(f_{[d+1:n]}(\theta_{t'}, z) - f_{[d+1:n]}(\theta_{t'}, z_{t'}(x)) = 0\right) d\omega_{x,t',\Delta t}(z) d\bar{\mu}(x)$$

$$\geq \int_{\mathcal{X}} (1 - \delta) d\bar{\mu}(x)$$

$$= 1 - \delta$$

(2) By definition, we have:

$$\int_{\mathcal{Z} \times \mathcal{X}} \mathbb{I}\left(f_{[d+1:n]}(\theta_{t'}, z) - f_{[d+1:n]}(\theta_{t'}, z_{t'}(x)) = 0\right) d\omega_{x,t',\Delta t}(z) d\bar{\mu}(x) \geq 1 - \delta$$

Denote the percentage of $x$ in the empirical distribution that the following equation holds by $p$:

$$\int_{\mathcal{Z}} \mathbb{I}\left(f_{[d+1:n]}(\theta_{t'}, z) - f_{[d+1:n]}(\theta_{t'}, z_{t'}(x)) = 0\right) d\omega_{x,t',\Delta t}(z) \geq 1 - q\delta$$

where $q > 1$. Hence, with probability $1 - p$, for $x \sim \bar{\mu}$,

$$\int_{\mathcal{Z}} \mathbb{I}\left(f_{[d+1:n]}(\theta_{t'}, z) - f_{[d+1:n]}(\theta_{t'}, z_{t'}(x)) = 0\right) d\omega_{x,t',\Delta t}(z) < 1 - q\delta$$

Since the indicator function is bounded above by 1, it is evident that:

$$\int_{\mathcal{Z} \times \mathcal{X}} \mathbb{I}\left(f_{[d+1:n]}(\theta_{t'}, z) - f_{[d+1:n]}(\theta_{t'}, z_{t'}(x)) = 0\right) d\omega_{x,t',\Delta t}(z) d\bar{\mu}(x)$$

$$< (1 - p)(1 - q\delta) + p = 1 - q\delta(1 - p)$$

We have:

$$1 - \delta < 1 - q\delta(1 - p)$$

$$1 > q(1 - p)$$

$$p > 1 - \frac{1}{q}$$

$\square$

## B CONNECTIONS BETWEEN THE FEATURE DYNAMICS PROPERTIES AND GENERALIZATION

Building upon the recognition of the memory and transferability phenomena, we can employ these concepts to describe their relationship with generalization. Specifically, given a data point $x$, epoch $t'$, and interval $\Delta t$, the quantity $\omega_{x,t',\Delta t}(z)$ can be interpreted as the measure induced by the shallower networks through data augmentation applied to a single input. Accordingly, for the entire empirical distribution, the expression

$$\Omega_{t',\Delta t}^{(i)} =: \int_{\mathcal{X}} d\omega_{x,t',\Delta t}(z) d\bar{\mu}_i(x)$$

represents the augmented data obtained by applying such transformations to the class $i$ of full training set via the shallower network. According to the memory and transferability phenomena, the deeper layers of the network are capable of effectively classifying the latent representations induced by this measure. Therefore, if the distribution induced by the true data distribution in the latent space is sufficiently close to this augmented distribution, the network is expected to generalize effectively.

**Theorem 3.** *If empirical-level memory holds at $(\delta, \Delta t)$ and epoch $t'$, then the generalization gap of class $i$ is bounded by the difference between the induced measure $f_{[1:d]\#}(\mu_i)$ and the empirical level $z_t$ distribution $\int_X d\omega_{x,t',\Delta t}(z)d\bar{\mu}_i(x)$.*

$$\text{Generalization Gap of label } i \leq TV\left(\Omega^{(i)}_{t',\Delta t} \,\middle\|\, f_{[1:d](\theta_{t'},\cdot)\#}(\mu_i)\right) + \delta$$

*where $TV(\mu|\nu)$ represent the total variation distance between $\mu$ and $\nu$ and the generalization gap of label $i$ is defined by:*

$$\left|\int_{\mathcal{X}} \mathbb{I}(f(\theta_{t'}, x) \neq i)\, d\mu_i - \int_{\mathcal{X}} \mathbb{I}(f(\theta_{t'}, x) \neq i)\, d\bar{\mu}_i\right|$$

*This gap reflects the difference between the test accuracy and the training accuracy for samples with label $i$.*

**Proof:**

$$\left|\int_{\mathcal{X}} \mathbb{I}(f(\theta_{t'}, x) \neq i)\, d\mu_i - \int_{\mathcal{X}} \mathbb{I}(f(\theta_{t'}, x) \neq i)\, d\bar{\mu}_i\right|$$

$$= \left|\int_{\mathcal{Z}} \mathbb{I}\left(f_{[d+1:n]}(\theta_{t'}, z) \neq i\right) df_{[1:d](\theta_{t'},\cdot)\#}(\mu_i) - \int_{\mathcal{Z}} \mathbb{I}\left(f_{[d+1:n]}(\theta_{t'}, z) \neq i\right) df_{[1:d](\theta_{t'},\cdot)\#}(\bar{\mu}_i)\right|$$

$$\leq \left|\int_{\mathcal{Z}} \mathbb{I}\left(f_{[d+1:n]}(\theta_{t'}, z) \neq i\right) df_{[1:d](\theta_{t'},\cdot)\#}(\mu_i) - \int_{\mathcal{Z}} \mathbb{I}\left(f_{[d+1:n]}(\theta_{t'}, z) \neq i\right)\left(\int_{\mathcal{X}} d\omega_{x,t',\Delta t}(z)d\bar{\mu}_i(x)\right)\right| \quad (1)$$

$$+ \left|\int_{\mathcal{Z}} \mathbb{I}\left(f_{[d+1:n]}(\theta_{t'}, z) \neq i\right)\left(\int_{\mathcal{X}} d\omega_{x,t',\Delta t}(z)d\bar{\mu}_i(x)\right) - \int_{\mathcal{Z}} \mathbb{I}\left(f_{[d+1:n]}(\theta_{t'}, z) \neq i\right) df_{[1:d](\theta_{t'},\cdot)\#}(\bar{\mu}_i)\right| \quad (2)$$

By the property of the total variation distance:

$$TV(\mu\|\nu) = \sup_{f \in L^\infty(\mu,\nu), |f| \leq 1} \int f d\mu - \int f d\nu$$

$(1)$ is upper bounded:

$$(1) \leq TV\left(\Omega^{(i)}_{t',\Delta t} \,\middle\|\, f_{[1:d](\theta_{t'},\cdot)\#}(\mu_i)\right)$$

$(2)$ is upper bounded by $\delta$ since the emperical-level memory holds:

$$\left|\int_{\mathcal{Z}} \mathbb{I}\left(f_{[d+1:n]}(\theta_{t'}, z) \neq i\right) \int_{\mathcal{X}} d\omega_{x,t',\Delta t}(z)d\bar{\mu}_i(x) - \int_{\mathcal{Z}} \mathbb{I}\left(f_{[d+1:n]}(\theta_{t'}, z) \neq i\right) df_{[1:d](\theta_{t'},\cdot)\#}(\bar{\mu}_i)\right|$$

$$= \left|\int_{\mathcal{Z}\times\mathcal{X}} \mathbb{I}\left(f_{[d+1:n]}(\theta_{t'}, z) \neq i\right) d\omega_{x,t',\Delta t}(z)d\bar{\mu}_i(x) - \int_{\mathcal{X}} \mathbb{I}\left(f_{[d+1:n]}(\theta_{t'}, z_{t'}(x)) \neq i\right) d\bar{\mu}_i\right|$$

$$\leq \int_{\mathcal{Z}\times\mathcal{X}} \left|\mathbb{I}\left(f_{[d+1:n]}(\theta_{t'}, z) \neq i\right) - \mathbb{I}\left(f_{[d+1:n]}(\theta_{t'}, z_{t'}(x)) \neq i\right)\right| d\omega_{x,t',\Delta t}(z)d\bar{\mu}_i(x)$$

$$\underset{(i)}{\leq} \int_{\mathcal{Z}\times\mathcal{X}} 1 - \mathbb{I}\left(f_{[d+1:n]}(\theta_{t'}, z) - f_{[d+1:n]}(\theta_{t'}, z_{t'}(x)) = 0\right) d\omega_{x,t',\Delta t_2}(z)d\bar{\mu}_i(x)$$

$$\leq \delta$$

Inequality (i) holds since

$$\left|\mathbb{I}\left(f_{[d+1:n]}(\theta_{t'}, z) \neq i\right) - \mathbb{I}\left(f_{[d+1:n]}(\theta_{t'}, z_{t'}(x)) \neq i\right)\right|$$
$$\leq 1 - \mathbb{I}\left(f_{[d+1:n]}(\theta_{t'}, z) = f_{[d+1:n]}(\theta_{t'}, z_{t'}(x))\right)$$

$1 - \mathbb{I}(\cdot) \geq 0$, so when $\left|\mathbb{I}\left(f_{[d+1:n]}(\theta_{t'}, z) \neq i\right) - \mathbb{I}\left(f_{[d+1:n]}(\theta_{t'}, z_{t'}(x)) \neq i\right)\right| = 0$, inequality holds. When $\left|\mathbb{I}\left(f_{[d+1:n]}(\theta_{t'}, z) \neq i\right) - \mathbb{I}\left(f_{[d+1:n]}(\theta_{t'}, z_{t'}(x)) \neq i\right)\right| = 1$, only one of the following equation holds:

$$f_{[d+1:n]}(\theta_{t'}, z_{t'}(x)) = i$$
$$f_{[d+1:n]}(\theta_{t'}, z) = i$$

Hence,

$$1 - \mathbb{I}\left(f_{[d+1:n]}\left(\theta_{t'}, z\right) - f_{[d+1:n]}\left(\theta_{t'}, z_{t'}(x)\right) = 0\right) = 1$$

the inequality holds.

□

Building upon the conditions outlined in Theorem 3, we provide a comprehensive generalization bound:

$$\text{Generalization Gap} = \left|\int_{\mathcal{X}\times\mathcal{Y}} \mathbb{I}\left(f(\theta, x) \neq i\right) d\mu_i d\nu(i) - \int \mathbb{I}\left(f(\theta, x) \neq i\right) d\bar{\mu}_i d\bar{\nu}(i)\right|$$

where $\nu$ is the true distribution of label and $\bar{\nu}$ is the empirical distribution of labels. We have:

$$\left|\int_{\mathcal{X}\times\mathcal{Y}} \mathbb{I}\left(f(\theta_{t'}, x) \neq i\right) d\mu_i d\nu(i) - \int \mathbb{I}\left(f(\theta_{t'}, x) \neq i\right) d\bar{\mu}_i d\bar{\nu}(i)\right|$$

$$\leq \left|\int_{\mathcal{X}\times\mathcal{Y}} \mathbb{I}\left(f(\theta_{t'}, x) \neq i\right) d\mu_i d\nu(i) - \int \mathbb{I}\left(f(\theta_{t'}, x) \neq i\right) d\bar{\mu}_i d\nu(i)\right| \quad (3)$$

$$+ \left|\int_{\mathcal{X}\times\mathcal{Y}} \mathbb{I}\left(f(\theta_{t'}, x) \neq i\right) d\bar{\mu}_i d\nu(i) - \int \mathbb{I}\left(f(\theta_{t'}, x) \neq i\right) d\bar{\mu}_i d\bar{\nu}(i)\right| \quad (4)$$

(3) is upper bounded by Thm2:

$$(3) \leq \sum_i \nu(i) TV\left(\int_{\mathcal{X}} d\omega_{x,t',\Delta t}(z) d\bar{\mu}(x) \,\middle\|\, f_{[1:d]}(\theta_{t'}, \cdot)_{\#}(\mu_i)\right) + \delta$$

(4) can be upper bounded by the total variation distance:

$$(4) \leq TV(\nu\|\bar{\nu})$$

Also, it can be upper bounded by the maximum probability of misclassification for empirical distribution of class $i$:

$$(4) \leq \left|\int_{\mathcal{X}\times\mathcal{Y}} \mathbb{I}\left(f(\theta_{t'}, x) \neq i\right) d\bar{\mu}_i d\nu(i)\right| + \left|\int \mathbb{I}\left(f(\theta_{t'}, x) \neq i\right) d\bar{\mu}_i d\bar{\nu}(i)\right|$$

$$\leq 2\sup_i \int_{\mathcal{X}} \mathbb{I}\left(f(\theta_{t'}, x) \neq i\right) d\bar{\mu}_i$$

Specifically, it quantifies the worst-case misclassification rate for each class $i$, multiplied by two, and then finds the supremum over all classes. It reflects whether the network has been well-trained.

# H    EXPERIMENTAL DETAILS

We conducted our experiments using PyTorch. All experiments were carried out on a server cluster equipped with NVIDIA GeForce RTX 4090 GPUs. Due to the relatively small size of the models used in the experiments, training can be completed within a few hours.

**Experiment 1 and 2**    Our experimental setup follows the most classic neural network experiment configuration, using the open-source code released under the MIT License [Aaron Chen, 2017] and BSD-2-Clause License [Yerlan Idelbayev, 2018].    For detailed information, please refer to https://github.com/aaron-xichen/pytorch-playground and https://github.com/akamaster/pytorch_resnet_cifar10. We acknowledge the original authors and have respected all licensing terms.

The density of consistency in the figures is estimated using kernel density estimation.

See Table 2 for the basic training settings. Additionally, the learning rate was reduced at specific epochs during training. For more detailed settings, see the URL provided earlier. Refer to Figure 19 for the precision of the loss curve during training.

Table 2: Training Settings Across Different Datasets

| Dataset | Optimizer | Batch Size | Learning Rate | Weight Decay | Momentum |
|---------|-----------|------------|---------------|--------------|----------|
| SVHN | Adam | 200 | 0.001 | 0.00 | – |
| STL10 | Adam | 200 | 0.001 | 0.001 | – |
| MNIST | SGD | 200 | 0.01 | 0.0001 | 0.9 |
| CIFAR10/100 | SGD | 128 | 0.1 | 0.0001 | 0.9 |

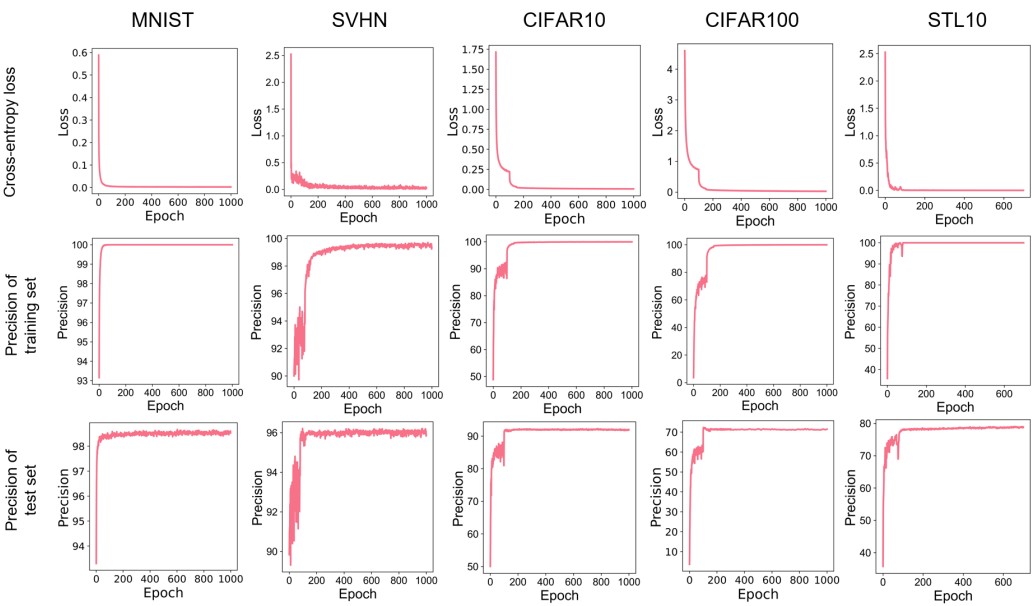

Figure 19: Loss and precision curve of training set and test set during training.

**Experiment 3**    In the experiments, the learning rate was set to $10^{-1}$ and the weight decay to $10^{-5}$, with training conducted for 200 epochs. The model was a multilayer perceptron (MLP) with layer widths of 28×28, 100, 2, and 10, respectively, and ReLU was used as the activation function. Notably, no activation function was applied after the penultimate layer. In the experiments, we visualized the features in the two-dimensional latent space.

