# OpenReview forum: "Feature Dynamics as Implicit Data Augmentation: A Depth-Decomposed View on Deep Neural Network Generalization"
_ICLR.cc/2026/Conference — Submitted to ICLR 2026_

### Official Review · Reviewer_MAff · 2025-10-29

**Soundness:** 2
**Presentation:** 2
**Contribution:** 3
**Rating:** 4
**Confidence:** 4

**Summary:**

This paper proposes a depth-decomposed lens on generalization: composite networks formed by mixing shallow layers from earlier epochs with deep layers from later epochs show strong temporal consistency. The authors interpret this as an implicit, structured form of data augmentation induced by training dynamics. They demonstrate the effect across CIFAR-10/100, SVHN, STL-10, and corrupted variants, show it collapses under random labels, and link it mechanistically to anisotropic SGD noise. A conceptual bound using temporally augmented feature distributions is also introduced.

**Strengths:**

1. Novel and intuitive framing: “feature dynamics as implicit augmentation.”
2. Empirical phenomena (memory, transferability, induction) are consistently reproduced on CIFAR, SVHN, STL-10, MNIST.
3. Strong controls: corrupted inputs vs. random labels.
4. Careful perturbation analysis linking SGD anisotropy to feature drift.
5. Theoretical bridge, even if informal, connecting dynamics to generalization.

**Weaknesses:**

1. Theory is conceptual only: TV bound is elegant but not computable, surrogates (MMD/Wasserstein) are not tested.
2. Scale: Evidence is limited to small/mid-scale datasets, no ImageNet or large transformer experiments.
3. BatchNorm handling in composites is not described and may confound results.
4. Optimizer sensitivity not analyzed, anisotropy is only shown for SGD though Adam is also used.
5. Relation to prior stitching/ensembling, no clear explanation on how their method differs from SWA/SWAG/temporal ensembling.

**Questions:**

1. How exactly is BatchNorm handled when combining shallow t1and deep t2?
2. Does temporal consistency hold under Adam and different batch sizes?
3. Can you show results on at least one larger-scale model?
4. Can you instantiate an MMD/Wasserstein surrogate for the TV bound?
5. How does this differ empirically from SWA/SWAG or temporal ensembling?

---

### Official Review · Reviewer_epy9 · 2025-11-01

**Soundness:** 1
**Presentation:** 1
**Contribution:** 1
**Rating:** 0
**Confidence:** 5

**Summary:**

This paper investigates the evolution of internal features during training. The authors observe that these features exhibit temporal consistency, and they interpret this phenomenon as a form of implicit data augmentation.

**Strengths:**

This work introduce interesting viewpoint on internal feature dynamics and the authors investigate their claims using multiple perspectives, including both empirical experiments and theoretical discussions.

**Weaknesses:**

### W1 Formatting and presentation errors
The manuscript suffers from a high volume of formatting and presentation errors which significantly hinder readability. For example:
* The terminology used to describe different sections of a single network is vague and potentially misleading. For example, using the expressions "shallow network" and "deep network" (e.g., line 47) to represent two parts within the same network can cause confusion. Readers may misunderstand this as a comparison between two independent networks with different depths. Using more standard and precise terms, such as the "early part" and "late part" of the network (or "lower/upper layers"), would significantly improve the paper's clarity.
* The paper uses several abbreviations before they are formally defined, which is confusing for the reader (e.g., "TV" and "MMD" are used in lines 67-69).
* The citation format is often incorrect or inconsistent. For instance, the citation in lines 70-72 should be corrected to a proper parenthetical style (e.g., "...transformers (Aubry et al., 2025)").
* Section 2 consists of only a single subsection, which is unnecessary and should be merged into the main section body.
* "Transferability" section title is introduced twice (consecutively as Section 4.3 and Section 4.4)
* Incorrect or confusing references to figures and sections.
    * Line 262: Appendix Figure 11-> Figure 11
    * Line 320: Appendix 14 -> Figure 14
    * Line 349 Appendix Fig.16 -> Figure 16
    * Line 406 Appendix Figure 15 -> Figure 15
* Line 126: $\theta_t$ appears before it is defined.

### W2 Weak logical connection between the claim and evidences
* I am not convinced by the argument in lines 159-161, where the authors claim that SGD stochasticity acts as implicit data augmentation. The paper states that due to SGD, "the hidden feature $z_t(x)$ does not remain a single deterministic point but evolves into a distribution". This statement seems incorrect in the context of this work, which analyzes the feature dynamics within a single network instead of considering multiple networks obtained by different randomness. Could the authors clarify this?
* The experimental results presented in Section 4 are conducted on CIFAR-10 training set and I believe these results are trivial artifact of model convergence. To provide meaningful evidence, the authors must conduct these same experiments on the test set. While the authors briefly attempt to argue that this is not a convergence artifact (lines 259-264), this discussion is not emphasized and is insufficient to address this major concern.
* In line 269-286, the authors provide discussion on NTK regime. However, it is unclear how this discussion is related to the main claim.

**Questions:**

See Weaknesses.

---

### Official Review · Reviewer_88De · 2025-11-01

**Soundness:** 4
**Presentation:** 4
**Contribution:** 3
**Rating:** 4
**Confidence:** 3

**Summary:**

The paper studies the problem of DNN generalisation, which cannot be effectively explained by classical ML tools. The framework developed by the authors focuses on the evolution of hidden representations and their role in network implicit auto-regularisation behaviour. The paper verifies the validity of the claims with a set of experiments on image classification tasks using popular datasets (MNIST, CIFAR, SVHN) and provides a link between SGD-induced noise to network generalisation.

**Strengths:**

1. The topic studied by the paper is of great importance to the community, and the angle at which the authors study the problem is relatively underexplored within the community; thus, the contributions of the work might be of great importance for the field.
2. The presentation is clear, the text is well structured, and the explanations are easy to follow.
3. Experiments are carefully designed, and the results support the claims made by the authors.

**Weaknesses:**

1. While the experiments are well-designed, the scope of the experiments is rather limited and therefore creates a question whether these observations would hold on bigger datasets or other architectures, let alone different domains (e.g. NLP).
2. The analysis does not study the effect of hyperparameter selection on the observed phenomena leaving the reader with a feeling that these observations should be regularly observed regardless of the hyperparameters. I don't know if that's authors intention but it's not clarified within the main text how broadly this phenomenon applies.
3. While the idea of studying hidden representation dynamics and their role in generalisation is interesting and underexplored, several works have studied this topic earlier and the authors havent compared their results nor provided any discussion. Specifically, The Tunnel Effect [1] shows that the networks naturally split into two parts: the extractor and the tunnel. The former acts as a feature extractor, while the latter acts as a classifier. Importantly, the authors in [1] showed a variant of the consistency experiments in the OOD scenario, a discussion with these results would benefit from locating this work within the current knowledge. Later [2] challenged the Tunnel Hypothesis by showing its dependence on the choice of hyperparameters, datasets (specifically, the tunnel got shorter or even disappeared on the datasets with higher image resolution) and the strength of augmentations. Thus, I'm slightly worried that the lack of hyperparameters sweeps does not allow us to judge how broadly these observations apply.


[1] https://arxiv.org/abs/2305.19753
[2] https://arxiv.org/abs/2405.15018

**Questions:**

My main question for the authors would be to provide an in-depth discussion of the works I have provided (and the related works around the topic) to better locate their findings (which I still find novel and interesting) within the field.

It would be ideal if the authors could run additional experiments to check whether their observations of self-consistency hold under different hyperparameters (e.g. higher image resolutions,) but I understand the short discussion time for this, so lack of these experiments won't be a decisive factor for my final recommendation.

---

### Official Review · Reviewer_qBci · 2025-11-01

**Soundness:** 3
**Presentation:** 3
**Contribution:** 3
**Rating:** 6
**Confidence:** 2

**Summary:**

This paper presents an interesting view of the dynamics of internal representations as data augmentations to explain the generalization ability of neural networks. The paper presents several claims on memory, transferability, and induction and provides experimental validation that explains these phenomena under the proposed framework.

**Strengths:**

**Provides new and interesting observations**:
Looking at generalization through the lens of composite networks is novel and interesting. In particular, the note that the forgetting phenomenon observed cannot be explained by the standard NTK assumption is a novel insight.

**Strong Experimental validation**:
All the claims presented in the work are empirically verified on a broad range of architectures and models, which provide evidence for the claims. The paper provides a thorough experimental evaluation of all the observed phenomena.

**Well Motivated Problem**:
Classical Learning theory has failed to explain generalization in neural nets, whereas the present work shows promise in its approach. The proofs presented in the paper are well structured and are

**Weaknesses:**

**Not prescriptive**:
This work raises several interesting observations that help explain the generalization properties of neural networks. However, it does not prescribe any new training algorithms that may be built from this framework that can improve generalization characteristics.

The paper provides details on reproducing experiments, including hyperparameters, system requirements, and setup. However, they do not provide the actual code used to run these experiments. Releasing the source code would greatly enhance reproducibility and trust.

**Questions:**

- Analysis of varying $d$: The paper would benefit from a discussion of varying values of $d$ compared to $n$ and its effects. In its current form, the paper makes no comments regarding the impact of the depth of consideration and how these effects are observed at varying depths.
- Could the authors define the term Temporal Augmentation more precisely?
- Could the authors formally state the ''Data Augmentation hypothesis''?
- Could the authors provide additional discussions on the formal definitions of memory, forgetting, and transferability? A remark condensing Definition 2 from Appendix G.A would greatly enhance readability.
- In lines 422-429, it is not clear from the notation where the source of randomness in the distribution of $\omega$ is present. Is it from the data, sampling of time instant, or both?
- The paper should add a section discussing the limitations of the work.

---

Minor Nitpicks (Did not affect decision):

- Several citations are not cited correctly, such as in lines 70, 71, 117, and 148. These can be resolved with the correct use of \citet and \citep
- Stating lines 100, 107, and 115 as \paragraph would enhance the readability of the work
- The section title for Section 4.3 is repeated
- Changing the scale in Figure 5 would enhance readability

I currently rate this paper a 6 with low confidence because I have not verified the proofs presented in detail, but I am looking forward to the discussion period and am open to strengthening the work.

---

> ### Comment · Reviewer_qBci · 2025-11-13
> **clarification of incomplete sentence**
>
> Dear Authors,
>
> Please read the last sentence of the third point in the Strengths section as:
> ```
> The proofs presented in the paper are well structured.
> ```
>
> My apologies for the inconvenience.

---

### Meta-Review · Area_Chair_HjNP · 2026-01-01

**Summary:**

This paper proposes an interesting lens on generalization: hidden-representation temporal consistency across training (via composite/stitching-style networks mixing early/late layers from different epochs) as an implicit form of data augmentation. Reviewers appreciated the novelty of the framing and the breadth of small-to-mid-scale vision experiments with thoughtful controls (e.g., corrupted inputs vs. random labels), and some found the mechanistic link to anisotropic SGD noise promising.

However, the current evidence and presentation fall short of the bar of acceptance. Key concepts (e.g., “temporal augmentation,” and “data augmentation hypothesis,” and notions of memory/forgetting) are not defined precisely, and several claims have unclear logical support (including ambiguity about sources of randomness and concerns about training-set/convergence artifacts). The evaluation is limited in scale and robustness (no large-scale models; insufficient hyperparameter/optimizer sensitivity), and the paper lacks sufficient discussion of the related works (e.g., tunnel-effect results, temporal ensembling/SWA/SWAG). Reproducibility is also weakened by the absence of released code and significant formatting/citation issues.

**Reviewer Concerns:**

The current evidence and presentation fall short of the bar of acceptance. Key concepts (e.g., “temporal augmentation,” and “data augmentation hypothesis,” and notions of memory/forgetting) are not defined precisely, and several claims have unclear logical support (including ambiguity about sources of randomness and concerns about training-set/convergence artifacts). The evaluation is limited in scale and robustness (no large-scale models; insufficient hyperparameter/optimizer sensitivity), and the paper lacks sufficient discussion of the related works (e.g., tunnel-effect results, temporal ensembling/SWA/SWAG). Reproducibility is also weakened by the absence of released code and significant formatting/citation issues.

The authors have not provided any rebuttal.

**Reviewer Scores:**

Due to the authors' lack of response, the reviewers' scores will remain unchanged.

---

### Decision · Program_Chairs · 2026-01-26

Reject